# Stand Structure Extraction and Analysis of *Camellia taliensis* Communities in Qianjiazhai, Ailao Mountain, China, Based on Backpack Laser Scanning

**DOI:** 10.3390/plants14162485

**Published:** 2025-08-11

**Authors:** Xiongfu Gao, Xiaoqing Shi, Weiheng Xu, Zengquan Lan, Juxiang He, Huan Wang, Leiguang Wang, Ning Lu, Guanglong Ou

**Affiliations:** 1College of Big Data and Intelligent Engineering, Southwest Forestry University, Kunming 650233, China; gxf@swfu.edu.cn (X.G.); xqshi123@swfu.edu.cn (X.S.); hejuxiang@swfu.edu.cn (J.H.); wanghuan@swfu.edu.cn (H.W.); ninglu@swfu.edu.cn (N.L.); 2Anciant Tea Plant Research Center, Southwest Forestry University, Kunming 650223, China; Lanzengquan@tsinghua.org.cn; 3College of Landscape and Horticulture, Southwest Forestry University, Kunming 650233, China; leiguanwang@swfu.edu.cn; 4College of Forestry, Southwest Forestry University, Kunming 650233, China; olg2007621@126.com

**Keywords:** ancient tea tree community, backpack laser scanning, stand spatial structure, stand non-spatial structure, species diversity

## Abstract

The stand structure of ancient tea tree (*Camellia taliensis*) communities is critical for maintaining their structural and functional stability. Therefore, this study employed backpack laser scanning (BLS) technology to extract individual tree parameters (diameter at breast height, tree height, relative coordinates, etc.) in seven sample plots (25 m × 25 m each) to analyze their spatial and non-spatial structure characteristics. Firstly, the accuracy of diameter at breast height (DBH) and tree height (TH) estimations using BLS resulted in a root mean square error (*RMSE*) of 4.247 cm and 2.736 m and a coefficient of determination (*R*^2^) of 0.948 and 0.614, respectively. Secondly, in this community, trees exhibited an aggregated spatial distribution (average uniform angle > 0.59), with small differences in DBH among adjacent trees (average dominance > 0.48) and a high proportion of adjacent trees belonging to different species (average mingling > 0.64). Ancient tea trees in the 5–15 cm diameter class face considerable competitive pressure, with values ranging from 14.28 to 179.03. Thirdly, this community exhibits rich species composition (more than 7 families, 8 genera, and 10 species, respectively), strong regeneration capacity (with an inverse J-shaped diameter distribution), uniform species distribution (Pielou evenness index > 0.71), and high species diversity (with a Shannon–Wiener diversity index ranging from 1.65 to 2.47 and a Simpson diversity index ranging from 0.71 to 0.91), and the ancient tea trees maintain a prominent dominant status and important value ranging from 19.36% to 49%. The results indicate that, under the current conditions, the structure and function of this community collectively exhibit relatively stable characteristics. BLS provides a powerful tool for the research and conservation of rare and endangered species.

## 1. Introduction

The ancient tea tree (*Camellia taliensis*) is a rare tea tree germplasm resource with important ecological, conservation, and economic value [1]. The stand structure of the ancient tea trees affect the stability and functions of its ecosystems [2]. To date, previous research on ancient tea trees has predominantly focused on resource surveys [3], molecular marker development [4], biochemical composition detection [5], and gene diversity [6], while relatively little research on the stand structure of ancient tea tree communities. Ancient tea trees are mainly distributed in the Yunnan, Guizhou, Sichuan, Chongqing, and Guangxi provinces in China [6]. Among them, Yunnan is the region with the richest ancient tea tree habitats. The ancient tea tree communities of Qianjiazhai, located in Zhenyuan County, Yunnan Province, are the largest and best-preserved ancient tea tree communities discovered to date [7,8], and Qianjiazhai is the most representative area for stand structure research. Therefore, it is crucial to choose the appropriate method during the investigation to minimize interference with the environment.

However, traditional forestry surveys have primarily relied on field-based investigation methods [9]. Typically, researchers establish fixed-area sample plots in the field and use appropriate instruments to manually measure each tree, recording individual parameters including diameter at breast height (DBH), tree height (TH), crown width (CW), and height to the first branch. This method is time-consuming and labor-intensive in large-scale forest surveys; moreover, it may also cause interference with the ecological environment and is especially detrimental to the protection of rare plant resources, such as ancient tea trees. In the past twenty years, light detection and ranging (LiDAR) technology has played an increasingly important role in forestry surveys by virtue of its active remote sensing characteristics [10,11]. Compared with traditional optical remote sensing, the point cloud data collected by LiDAR can obtain accurate individual tree structure parameters, including TH, DBH, CW, volume, and coordinates, which can be used to calculate stand structure characteristics and extract quantitative structure models [12,13]. According to different platforms, LiDAR systems can be classified into four types, which include terrestrial laser scanning (TLS) [14], mobile laser scanning (MLS) [15], airborne LiDAR [16], and spaceborne LiDAR [17]. Since the 1990s, spaceborne LiDAR technology has matured, relying on satellites or the International Space Station to enable large-scale three-dimensional data acquisition for long-term dynamic monitoring [17]. But spaceborne LiDAR exhibits limited spatial resolution, which hinders its ability to capture fine surface details, and its sparse point cloud density reduces data accuracy [18]. Airborne LiDAR employs a top-down data collection approach that enables efficient coverage of large areas and accurate measurement of canopy structure, but this approach hinders its ability to capture detailed information about understory vegetation and thus restricts its potential for use in understory individual-tree studies [19]. TLS is characterized by high measurement accuracy and spatial resolution [15,20]. Due to the greater workload, TLS is used for single objects or relatively small areas [21], such as natural monuments (single trees) [22] or landslides (in geomorphological studies) [23]. Compared with TLS, MLS has greater flexibility and efficiency and is widely used in forest inventory and management. BLS, as a type of MLS, has been widely used in forest surveys, including estimating the aboveground biomass (AGB) of a *Picea crassifolia* forest [24], optimizing the spatial structure of a *Metasequoia glyptostroboides* plantation forest [25], measuring and locating trees in natural forests [15], and extracting the DBH and TH of individual trees [26,27]. These studies indicate that BLS can provide more feasible and efficient technical support for detailed investigations. Therefore, applying it to the ancient tea community of Qianjiazhai with complex terrain to verify its efficiency and performance is worth exploring.

Stand structure refers to the architectural and functional elements that constitute a forest and that reflect the regeneration mechanism, competition pattern, and self-regulation ability of the community [28,29,30]. Stand structure includes the spatial and non-spatial structure of the forest [31,32]. The spatial structure describes the relative positional relationship between trees in space, which determines the competition trend and niche distribution among individuals and also affects the health status, growth potential, and stability of stands [33]. There are many indices used to quantify and describe the spatial structure, such as the Clark and Evans index [34], Ripley’s K-function [35], the O-ring statistic [36], uniform angle index [37], mingling [37], dominance [38], and the Hegyi index [39]. Of these, the uniform angle index, mingling, dominance, and the Hegyi index have been widely used in stand structure analysis, as they respectively describe the spatial distribution pattern, extent of mixing, tree size, and competitive intensity among trees. In recent years, researchers have used the above spatial structure indicators to analyze the evolution of stand spatial structure during the expansion of Moso bamboo [40] and explored the influence of spatial structure on Chinese fir (*Cunninghamia lanceolata*) plantations’ AGB [41], as well as the impacts of a natural *Quercus aliena var. acuteserrata* forest’s spatial structure under different forest management modes [29]. The purpose of these indicators is to reveal the health status, growth potential, and stability of the stands. Non-spatial structure focuses on the species composition and quantitative characteristics of communities, and it is an important dimension that reflects species diversity and community stability [42]. Commonly used indicators include the Patrick and Margalef richness index, which measures the richness of the number of species [43]; the Shannon–Wiener and Simpson diversity indices, which reflect the level of community diversity [44]; and the Pielou evenness index, which reflects the balance of species distribution in the stand [45]. Moreover, the importance value (IV) can be used as a measure of plant species dominance [45], while the diameter class structure reflects the successional stage of the community [46]. For example, the species diversity of *Betula alnoides* [47] and *Pinus yunnanensis* natural forests [48] in Dehong Prefecture, Yunnan Province, was quantified based on the aforementioned indicators, and both forest types exhibited relatively stable community structure and strong capacity for natural regeneration. Previous research has predominantly focused on common or major forest types, with a notable paucity of research on ancient tea tree communities. The ancient tea tree of Qianjiazhai is one of the national protected two-grade rare plants and an important germplasm resource bank [8]. It is of great value in the conservation of genetic resources, tea variety improvement, and the maintenance of ecosystem diversity. The stand structure of ancient tea tree communities greatly affects the growth status, population regeneration, and ecosystem stability of the ancient tea tree. Revealing the structural characteristics of ancient tea tree communities is one of the important ways for the sustainable protection and scientific management of ancient tea trees.

Overall, BLS could access the spatial and spatial structures of ancient tea tree communities, which could help us to explore their ecological functions and stabilization. The objectives of this study were to (a) verify the efficiency and performance of BLS in ancient tea trees; (b) quantify the stand structure of ancient tea tree communities; and (c) evaluate the stability of ancient tea tree communities based on spatial and non-spatial structures. The abbreviations used in our research are listed in Table A1.

## 2. Materials and Methods

### 2.1. Study Area

The study area is located in the core area of the Qianjiazhai National Nature Reserve in Ailao Mountain, Zhenyuan County (23°24′ N–24°22′ N, 100°21′ E–101°31′ E), Pu‘er City, Yunnan Province. The Qianjiazhai area covers approximately 762.4 km^2^, with ancient tea tree communities occupying over 1916.5 hm^2^, making it one of the most representative regions for wild ancient tea tree communities in the Ailao Mountains [8]. In the region, the average annual temperature is between 10 °C and 12 °C, precipitation is above 1500 mm, and the rainy season is mainly in July and August [49]. The area experiences minimal seasonal temperature variation, with a frost-free period lasting from April to October [2]. The annual relative humidity ranges from 78% to 88%. It belongs to the climate zone of the northern margin of the central subtropical zone, and its vegetation presents the mountain vertical zone vegetation type, with evergreen broad-leaved forests in the southern subtropical zone [2]. The ancient tea tree community is the dominant species in ecosystem. The ancient tea trees are mostly about 1000 years old and are distributed mainly at altitudes of 2100 to 2500 m. The study area and sample plot locations are shown in Figure 1.

### 2.2. Research Workflow Overview

The detailed workflow of this study is depicted in Figure 2. Firstly, according to the distribution of ancient tea trees in the study area, 7 representative sample plots of ancient tea tree communities were selected. We utilized BLS technology to collect point cloud data from 7 sample plots and collected the corresponding field data. Secondly, the point cloud data of the sample plots were pre-processed, including clipping, sampling, denoising, cloth simulation filtering (CSF), and elevation normalization. Thirdly, the point cloud data of each individual tree was segmented and extracted based on the LiDAR360 V 7.0 software. In addition, parameters were extracted from individual tree point clouds, including TH, DBH, and relative coordinates, and verified for TH and DBH extraction accuracy. Finally, the spatial structure characteristics and non-spatial structure characteristics were calculated and quantified by combining the BLS-extracted data and the field data.

### 2.3. Data Acquisition and Pre-Processing

#### 2.3.1. Field Data

The field data were collected in November 2023, and a total of 7 sample plots (25 m × 25 m each) were set up. The DBH, TH, relative coordinates, and species were recorded for each tree with DBH ≥ 5 cm. The DBH was measured using a diameter tape at a height of 1.3 m above the ground, while TH was determined using an altimeter. The tree species were identified by an experienced taxonomy expert and recorded via videos and pictures. However, the complex terrain and dense forest cover within the study area result in unstable Real-Time Kinematic Global Positioning System (RTK-GPS) and LiDAR-GPS (Light Detection and Ranging Global Positioning System) signals, thereby hindering the acquisition of absolute coordinates. Simultaneous Localization and Mapping (SLAM) technology enables precise estimation of individual tree locations in forest environments by generating spatial coordinates within a local reference frame, without relying on high-precision Global Navigation Satellite System (GNSS) systems, such as Real-Time Kinematic (RTK) [50]. Therefore, the positions of all trees were replaced by the relative coordinates obtained after the pre-processing BLS data. The basic information in each sample plot is shown in Table 1.

#### 2.3.2. The Acquisition and Pre-Processing of Backpack Laser Scanning Data

We used the LiBackpack DGC50H LiDAR system (Digital Green Valley Technology Co., Ltd., Beijing, China) to obtain point cloud data for each sample plot. The system is equipped with two laser sensors that rotate in the horizontal and vertical directions. The horizontal and vertical field angles were 360° and 180°, respectively, and the measurement range was within 100 m, where 640,000 laser pulses were emitted per second. During the data collection phase, the surveyor followed a double Z-shaped trajectory through the forest at a steady pace while carrying the BLS system to ensure the stability of the point cloud data.

To accurately extract the point cloud of individual trees and to increase the computational speed of the algorithm, a systematic series of data pre-processing steps were applied to the raw data. The actual point cloud data of each plot were obtained by precise clipping based on the location information of four marker points. A random sampling algorithm was utilized to sample the point cloud, thereby minimizing the impact of high data density on segmentation efficiency [51]. This algorithm accomplishes sampling by setting a threshold that retains 80%, where each point in the point cloud has the same probability of being sampled, and random sampling does not change the original positional distribution of the point cloud. The statistical outlier removal (SOR) algorithm was utilized to reduce the impact of noise on the segmentation accuracy of the sample point cloud, with the threshold parameter set to a domain composed of 10 points and a standard deviation of 2 [40]. On this basis, the CSF algorithm is applied to classify the ground and no-ground point clouds [52]. A digital elevation model (DEM) was generated from the filtered ground points using a triangulated irregular network (TIN) approach. The non-ground point cloud data were subsequently normalized based on the DEM to eliminate the influence of terrain undulations on height measurements [53]. The step of height normalization does not change the spatial distribution of trees and branches. These pre-processing steps are collaboratively accomplished by the point cloud magic (PCM, https://www.lidarcas.cn) and LiDAR360 (https://www.lidar360.com) software. The visualization of the BLS pre-processing workflow is shown in Figure 3.

### 2.4. Extraction of Individual Tree Parameters

Based on pre-processed point cloud data, the direct point-cloud-oriented Comparative Shortest Path (CSP) algorithm was used for point cloud segmentation to obtain the point cloud data of individual trees [54]. The algorithm uses the Density-Based Spatial Clustering of Applications with Noise (DBSCAN) algorithm for tree trunk detection [55]. Subsequently, it calculates the path distance from the tree crown point to the root of the tree trunk. Based on this distance, the crowns of different individual trees can be separated. Finally, combined with the segmentation results of each tree trunk and its corresponding tree crown, the point cloud of individual trees was obtained. However, a few individual trees were misdivided. These errors were corrected manually via visual interpretation using LiDAR 360 software. Tree parameters, including DBH, TH, and relative coordinates, were extracted from the segmented individual tree point clouds. Point cloud slices with a thickness of 10 cm (ranging from 1.25 to 1.35 m) were extracted from the segmented point cloud data, and they were converted into two-dimensional data. The least squares circle fitting method was applied to obtain the DBH of each individual tree [56,57]. TH was extracted as the absolute difference between the highest and lowest points in the vertical direction within each individual tree point cloud. The position of the center point of the root of each tree is selected as the relative coordinate (Figure 4). TH, DBH, and relative coordinate of individual trees are implemented using the Python 3.10.

### 2.5. Quantification of Stand Structure

#### 2.5.1. Spatial Structure Unit

The spatial structure unit is the most fundamental unit formed by a central tree and its closely adjacent trees within the stand [40]. It is crucial for understanding ecological relationships and assessing the structural functions of a forest stand. Voronoi diagrams were utilized to define spatial structural units within each sample plot, as illustrated in Figure 5. Each Voronoi polygon encompasses only a central tree, with the trees located in its neighboring Voronoi polygons regarded as adjacent trees [58]. Every pair of adjacent trees were connected by straight lines, creating a Delaunay triangulation. The length of each straight line is the distance between two adjacent trees. At the same time, the arccosine function is used to calculate the angle between the central tree and the adjacent trees. The Delaunay triangulation, Voronoi polygon construction, calculation of angles (using the arccosine function), and determination of the number of adjacent trees were all implemented using Python 3.10. These computations were performed based on the spatial coordinates of individual trees. In addition, to eliminate the edge effect, a rectangular buffer belt with a width of 2.5 m is generated at the plot edge. Trees located within the buffer zone are classified as neighboring trees, while trees located within the inner rectangle can be either central trees or neighboring trees. The parameters of each tree include DBH, species information, TH, number of neighboring trees, distance, and angle, which were used as the basic data for constructing the stand structure index. In this study, a central tree and its four closest neighboring trees were used to calculate the stand spatial structure index [59].

#### 2.5.2. Stand Structure Indices

The functions and processes of forest ecosystems are influenced by the stand spatial structure. In this study, the uniform angle index, mingling, dominance, and Hegyi index were used to quantify and analyze the spatial structure of ancient tea communities [37,39]. The uniform angle index describes the uniformity of adjacent trees around a central tree, reflecting the spatial distribution pattern of trees [37]. Mingling describes the extent of isolation between different species in the stand [37]. Dominance represents the differentiation in size among trees, reflecting the dominance of different trees in the stand [38]. The Hegyi index reflects the stress of competition between the center tree and its adjacent trees [39]. To effectively differentiate trees’ structural characteristics, these indices were classified into five levels with an interval of 0.25. Their calculation formulas are given below.

Uniform angle index (W):
(1)$$W_{i}=\frac{1}{4}\sum_{j=1}^{4}v_{ij}$$
where $W_{i}$ is the uniform angle index of the central tree; when the *j*-th angle α is less than the standard angle $\alpha_{o}$ = 72°, $v_{ij}$ = 1; otherwise, $v_{ij}$ = 0.

Mingling (M):
(2)$$M_{i}=\frac{1}{4}\sum_{j=1}^{4}v_{ij}$$
where $M_{i}$ is the diversity mingling of the central tree; when central tree *i* and adjacent tree *j* are different species, $v_{ij}$ = 1; otherwise, $v_{ij}$ = 0.

Dominance (U):
(3)$$U_{i}=\frac{1}{4}\sum_{j=1}^{4}v_{ij}$$
where $U_{i}$ is the neighborhood comparison of the central tree; when the DBH of the adjacent tree *j* is larger than the DBH of the central tree *i*, $v_{ij}$ = 1; otherwise, $v_{ij}$ = 0.

Hegyi index:
(4)$$CI_{i}\;=\;\sum_{j=1}^{4}\frac{d_{j}}{d_{i}L_{ij}}$$
(5)$$\begin{array}{c} CI=\sum_{i=1}^{N}CI_{i} \end{array}$$
where $CI_{i}$ is the competition index of the *i* tree; *CI* is the total competition index of a population or a class tree; $d_{i}$ is the DBH of the central tree; $d_{j}$ is the DBH of the competing tree *j*; $L_{ij}$ is the distance between the central tree and the competing tree; and *N* is the total number of trees.

In addition, non-spatial structure refers to metrics that are not associated with the spatial arrangement of trees, such as species composition, species diversity, importance value (IV), and diameter class structure. Species composition is characterized by family, genus, and species. Species diversity is evaluated by employing the Patrick richness index, Margalef richness index, Shannon–Wiener diversity index, Simpson dominance index (D), and Pielou evenness index (E) [43,44,45]. The IV is used to measure the dominance of tree species. The higher the Patrick richness index and Margalef richness index, the more diverse the species composition. The higher the Shannon–Wiener diversity index, the greater the species diversity within the stand. The higher the Simpson dominance index, the greater the species dominance. As the Pielou evenness index increases, the distribution of species individuals becomes more uniform. The higher the importance value, the higher the dominance of the species. Their calculation formulas are as follows:

Patrick richness index (R):
(6)R= S

Margalef richness index (MA):
(7)$$MA=\;\frac{S\;-\;1}{lnN}$$

Shannon–Wiener index (H):
(8)$$\begin{array}{c} H\;=\;-\sum_{i=1}^{s}p_{i}lnp_{i} \end{array}$$

Simpson index (D):
(9)$$\begin{array}{c} D\;=\;1\;-\;\sum_{i=1}^{S}p_{i}^{2} \end{array}$$

Pielou index (E):
(10)E= H/lnS

Importance values (IV):
(11)IV= (RA + RF + RD)/3× 100%
where *S* represents the number of species in the plant community, *N* denotes the total number of individuals of all species, $p_{i}$ represents the proportion of the number of individuals of the *i*-th species to the total number of individuals of all species, and *RA* (relative abundance) denotes the relative abundance of a species within a community, the percentage of the total number of the species to the total number of individuals in the stand. *RF* (relative frequency) signifies the relative frequency of a species, the percentage of frequency of occurrence of this species to the total frequency of all species in the stand. *RD* (relative dominance) represents the relative dominance of a species, represented by the percentage of the basal area at breast height of the species to the total basal area at breast height of the stand. The indices were computed using a combination of Python 3.10 and Microsoft Excel 2024.

### 2.6. Evaluation Metrics of DBH and H

In this study, the field measurement data (DBH and TH) were used to verify the accuracy of the DBH and TH derived by BLS. The coefficient of determination (*R*^2^), root mean square error (*RMSE*), relative root mean square error (*rRMSE*), and mean absolute error (*MAE*) were used to evaluate the accuracy of BLS-derived data. A higher *R*^2^ value indicates a stronger correlation between the field measurement data and the BLS-derived data. Smaller *RMSE*, *rRMSE*, and *MAE* values indicate higher prediction accuracy. *Bias* is used to determine the degree of overestimation or underestimation of estimates relative to field measurement data. *Bias* (%) represents the percentage deviation of estimates from field measurement data, quantifying relative overestimation or underestimation. All these indices were calculated using Python 3.10. The calculation formulas are as shown in Equations (12)–(17).
(12)R2=1−∑i=1n(yi−yi^)2∑i=1n(yi−y¯)2
(13)RMSE=1n∑i=1n(yi−yi^)2
(14)$$rRMSE=\frac{RMSE}{\bar{y}}\times 100$$
(15)MAE=∑i=1n|yi−yi^|n
(16)bias=1n∑i=1nyi−yi^
(17)$$bias\left(\%\right)=\frac{bias}{\bar{y}}\times 100$$
where $y_{i}$ is the field measurement data (DBH or TH) of the *i*-th individual tree; yi^ is the BLS extraction data (DBH or TH) of the *i*-th individual tree; *n* is the total number of trees in the sample plot; and $\bar{y}$ is the arithmetic of field measurement data (DBH or TH).

## 3. Results

### 3.1. Accuracy Evaluation of DBH and TH Extraction Using BLS

To evaluate the accuracy of DBH and TH extracted using BLS, we selected both field-measured data and BLS-extracted data from 535 trees. Six evaluation metrics were employed, including *R*^2^, *RMSE*, *MAE*, *rRMSE, Bias*, and *Bias* (%). We compared DBH and TH extracted using BLS data with the diameter at breast height and tree height of reference field measurements (Figure 6). As illustrated in the scatter plots, the coefficient of determination (*R*^2^ = 0.948) of the DBH extracted by BLS with field measurement was higher than that of TH (*R*^2^ = 0.614). The *RMSE, MAE*, *rRMSE*, *Bias*, and *Bias* (%) between the measured DBH and the DBH extracted from BLS were 4.247 cm, 3.383 cm, 19.51%, 1.366 cm, and 6.275%, respectively. The *RMSE*, *MAE*, *rRMSE*, *Bias*, and *Bias* (%) between the measured TH and the TH extracted from BLS were 2.736 m, 2.289 m, 25.67%, 2.245 m, and 21.069%, respectively. The results indicate that the accuracy of DBH derived from BLS is higher than that of TH, and BLS can be applied for stand structure analysis.

### 3.2. Stand Spatial Structure Characteristics

Figure 7 illustrates the distribution of W, U, and M values in different grades from Plot 1 to Plot 7. As can be seen, from Plot 1 to Plot 7, W = 0.75 (aggregation distribution), with values ranging between 37% and 53% (Figure 7a). Under the condition of W = 0.75, the proportion of Plot 2 and Plot 3 was highest (53%). The range of W = 0.5 (random distribution) was 23%–61%. When trees were randomly distributed, the proportion of Plot 1 was the highest (61%), and the proportion of Plot 5 was the lowest (23%). The results show that the spatial distribution of trees in sample plots was mainly characterized by random and low aggregation distribution. The distribution of U among different grades was relatively balanced (10%–37%), indicating that the DBH differences between the central tree and adjacent tree were small (Figure 7b). The proportion of trees with M = 0.75 (intense mingling distribution) and M = 1 (extremely intense mingling distribution) were relatively higher, ranging from 17% to 37% and 25% to 61%, respectively (Figure 7c). Under the condition of extremely intense mingling distribution (M = 1), the proportion of Plot 1 was the highest (61%). Under the condition of intense mingling distribution (M = 0.75), the proportion of Plot 1 was the lowest (17%). This indicates strong species isolation in all sample plots. The average values of W, U, and M exceeded 0.59, 0.48, and 0.64 in all sample plots, respectively (Figure 7d). Overall, all sample plots demonstrate low aggregate spatial distribution, uniform tree size, and strong species isolation.

Figure 8 shows the distribution of diameter classes and the competitive pressure of ancient tea trees in the seven sample plots. The results showed that the ancient tea tree diameter steps were mainly distributed in classes I and II. There was no diameter class I step in sample plot 1, and sample plot 3 had the most percent value for diameter class I, at 76%. However, there were more ancient tea trees with lower diameter classes distributed in the sample plots, while the competitive pressure was higher. Among these sample plots, the competitive pressure of the ancient tea tree community in sample plot 3 in diameter class I was the largest, with a total CI of 179, followed by the total CI of diameter class I in sample plot 4, which was 105.35. In addition, the figure demonstrates the gradual decrease in competitive pressure with increasing diameter class in the ancient tea tree community. The total CI of the largest diameter class of ancient tea trees in sample plot 4 was 0.37, while the total CI of the largest diameter class in sample plot 6 was only 6.21. Moreover, sample plot 3 had the highest average CI of 5.33 among all the sample plots, indicating that the competitive pressure was higher in these sample plots. Therefore, this study showed that the competitive pressure of the ancient tea community mainly originated from the ancient tea trees with smaller diameter classes, while the ancient tea trees with larger diameter classes were more competitive.

### 3.3. Stand Non-Spatial Structure Characteristics

The distribution of the five species diversity indices in all sample plots is shown in Figure 9. Patrick richness and Margalef richness values were more than 10 and 2.28, respectively. Plot 3 exhibited the highest species richness (Patrick richness = 28; Margalef = 5.28), while Plot 2 achieved the lowest (Patrick richness = 10; Margalef = 2.28), and other plots had relatively balanced values. The Shannon–Wiener index ranged from 1.65 to 2.47, with Plot 2 having the lowest value (1.65), while the values for the other plots were all above 2. All the Pielou index and Simpson index values were in close proximity to the theoretical maximum value of 1. The results indicate that the ancient tea tree community exhibits a generally high level of species diversity and a relatively stable community structure.

For brevity, Table 2 shows the distribution of the importance values of the three most frequent species in each individual sample plot. The detailed information is shown in Table A2. From Plot 1 to Plot 7, the relative abundance of ancient tea trees exceeded 21.57%. In Plot 2 and Plot 3, the relative abundance of ancient tea trees was the highest, reaching 51.92% and 42.17%, respectively. The relative dominance of the ancient tea trees surpassed that of other species, ranging from 17.46% to 53.42%. Similarly, in all sample plots, the number of ancient tea trees was high. The number of ancient tea trees was 27 in Plot 2, 70 in Plot 3, and 29 in Plot 4, respectively. The IV of the ancient tea trees (*Camellia taliensis*) was highest among all sample plots, ranging from 19.36% to 49.00%. These results indicate that the ancient tea tree is a dominant and abundant species.

Figure 10 shows the distribution of families, genera, and species in each plot. The numbers of families, genera, and species were greater than 7, 8, and 10 in all plots, respectively. Plot 3 was the most abundant, including 17 families, 24 genera, and 28 species. Plot 1 and Plot 2 had the lowest numbers, with only 8 families, 8 genera, and 11 species and 7 families, 9 genera, and 10 species, respectively. These results indicate that the plant species were abundant, and the species composition was relatively complex.

The distribution of diameter classes in all sample plots is shown in Figure 11. With the increase in diameter class, the number of trees decreased gradually. The total number of trees in diameter classes I and II was the highest, with the number of individuals ranging from 9 to 137. As the diameter class increased, the number of trees decreased stepwise, with all number of individuals remaining below 12. The diameter class distribution was characterized by a typical inverted J-shape in all sample plots. This suggests that the community possesses a strong capacity for self-renewal.

## 4. Discussion

### 4.1. Adaptability of Backpack Laser Scanning in the Study of the Stand Structure of Ancient Tea Tree Communities

High-efficiency and non-destructive methods are essential for the conservation of ancient tea trees. The Qianjiazhai area, an important nature reserve in China, has a complex and varied topography. Traditional manual investigation is time-consuming and labor-intensive, which limits its applicability for large-scale monitoring. In contrast, BLS is characterized by being non-destructive and high-efficiency, and it is widely used in forest surveys [15]. Previous research indicates that to collect point cloud data of a 10 m × 40 m sample, traditional measurement methods require 3–4 people to complete data collection, while BLS only requires one person [60]. In this study, the average time for BLS to collect data from each plot (25 m × 25 m) was approximately 15 min. Despite deploying a field team of six people, due to the rugged terrain of the study area, field data collection was limited to an average of 1–2 sample plots per day. For each plot, BLS data pre-processing and individual tree segmentation took approximately 15 min, whereas correcting segmentation errors (such as misdivided trees) required an additional average of 1–2 h. With a total of seven plots, the cumulative post-processing time amounted to roughly 10 h. Therefore, BLS is an efficient and convenient means to investigate ancient tea tree communities.

Accurately extracting the DBH of trees is crucial for forest inventory and management. The Hough transform, the RANSAC (Random Sample Consensus) algorithm, and the least squares circle fitting method are often used to extract the DBH [61,62,63]. Hough transform typically requires data rasterization, and this process may lead to the loss of detailed information [64]. The RANSAC algorithm demonstrates strong resilience to noise and outliers in complex environments. Because it is computationally intensive, the number of iterations increases considerably as data complexity and noise levels rise, leading to reduced computational efficiency [65]. In contrast, the least squares circle fitting method directly employs point cloud data for fitting without rasterization, thereby preserving more detailed information. In this study, the DBH extraction achieved high accuracy (*R*^2^ = 0.948; *RMSE* = 4.247 cm; *MAE* = 3.383 cm) compared to previous studies [66]. However, both the TH (*Bias* = 2.245 m; *Bias* (%) = 21.069%) and DBH (*Bias* = 1.366 cm; *Bias* (%) = 6.257%) of the ancient tea trees were underestimated. This may be due to the occlusion interference of BLS in penetrating the canopy and the complex stand structure affecting the integrity of vertical distribution and the accuracy of feature identification in the BLS point cloud [67]. However, the errors of TH extraction mainly depend on point cloud quality and forest types. Compared to results from a coniferous forest (with an *RMSE* ranging from 1.272 to 5.07 m) [68], in our study, the accuracy of TH generally showed a similar but slightly higher accuracy. BLS operates in a bottom-up fashion, where dense upper canopy layers may cause the laser to reflect multiple times as it passes through the foliage [69]. This can prevent some laser pulses from reaching the tree’s uppermost sections, resulting in point cloud data failing to capture the highest points [70]. In addition, the complex and dense forest structure in Qianjiazhai further hinders the penetration ability of BLS, leading to errors in TH.

### 4.2. The Stability of Ancient Tea Tree Community Structure and Function

The spatial structure of a stand determines the spatial distribution patterns and resource acquisition capabilities of plants within a community while also influencing growth dynamics, structural stability, and species diversity [71]. In general, the more stable the community structure, the more likely the spatial distribution of trees will be random or exhibit low aggregation [72]. In ancient tea community plots, the average uniform angle value ranged from 0.59 to 0.74 (Figure 7d), indicating a low level of aggregation. Such a distribution pattern promotes niche differentiation and species complementarity, thereby enhancing both structural stability and functional redundancy within the community [73]. The average dominance value of the ancient tea tree community plots, ranging from 0.48 to 0.66 (Figure 7d), reflects a balanced distribution between dominant and subordinate trees. This balance facilitates interspecific complementarity and efficient resource utilization, contributing to the functional stability of the community [74]. The average mingling value, ranging from 0.64 to 0.83 (Figure 7d), indicates high species mixing. This high mingling promotes interspecific complementarity and niche differentiation, reducing resource competition intensity and enhancing the dynamic stability of the community [75].

MacArthur [76] and Elton [77] proposed the diversity–stability hypothesis, which posits that species diversity positively influences community stability: the higher the species diversity, the greater the community’s resistance to external disturbances, and the more stable the community and ecosystem. The Shannon–Wiener diversity index (H > 1.65), Simpson diversity index (D > 0.71), and Pielou evenness index (E > 0.71) of ancient tea tree communities are significantly higher than those of typical Yunnan pine primary forests (H = 1.274, D = 0.635, E = 0.791) [78], indicating richer species composition, more even distribution, and greater functional stability. Tilman [79] posits that an increase in species diversity within a community enhances its resistance stability to external disturbances, thereby contributing to the maintenance of community stability. Gardner and Ashby [80] emphasize that higher species diversity reduces the strength of interspecific interactions among community members, consequently promoting community stability. The diameter class structure reflects the growth dynamics and regeneration potential of a community [46]. Previous studies have shown that undisturbed natural forests typically exhibit an inverse J-shape [81,82]. Qianjiazhai, located in the Ailao Mountains, is a region of natural forest within the national nature reserve. In ancient tea tree communities, the diameter class structure mainly exhibits this typical inverse J-shape, characterized by an abundance of small-diameter individuals, suggesting strong natural regeneration and sustained successional potential. However, any factor that disrupts the balance among regeneration, growth, and mortality can alter its characteristics. In particular, various natural disturbances often occur, such as drought, heavy winds and snow, landslides, floods, plant diseases and insect pests, and animal browsing [83,84,85]. Even minor disturbances can potentially shift the diameter distribution away from its theoretical equilibrium [46], which may explain why a reverse J-shape was not observed in Plot 1. Furthermore, both spatial and non-spatial structural characteristics indicate that ancient tea tree communities possess high structural stability and functional redundancy, demonstrating strong ecological adaptability and continuous successional capacity.

### 4.3. Strengths and Limitations

This study marks the first application of BLS technology to the conservation of ancient tea resources, offering a complementary alternative to traditional survey methods. While traditional surveys remain essential in forestry research, they often encounter challenges related to data accuracy and operational efficiency, especially in complex or inaccessible terrain [86]. In contrast, as a portable, non-contact three-dimensional scanning technology, BLS can offer efficient data acquisition and show potential adaptability to diverse and challenging field environments [15]. Nevertheless, the study has certain limitations. (1) Although the combined use of LiDAR360 and the PCM software enabled accurate segmentation of individual trees, the processes of denoising, filtering, and separating ground from non-ground points still depend on preset parameters. The generalizability of this method across diverse datasets requires further validation. While manual correction of segmentation errors (such as misdivided trees) remains feasible for small-scale studies, it can become increasingly time-consuming and inefficient when applied at larger scales. To address this, we propose two key improvements. First, the adoption of advanced segmentation algorithms, particularly deep learning models optimized for forest point cloud data, such as PointNet++ [87] and VoxelNet [88], significantly reduces the reliance on manual corrections. Second, automating repetitive post-processing tasks via batch scripting and modular toolchains can improve efficiency and reduce processing time. (2) Species identification currently depends primarily on recognition during field work. In the future, integrating reflection intensity data with deep learning models may significantly improve classification accuracy and improve investigation efficiency. (3) Finally, the current findings are based on a limited number of sample plots. In the future, studies will expand both the number and area of sampled plots to improve the comprehensiveness of the analysis.

## 5. Conclusions

In this study, we applied backpack laser scanning (BLS) to quantitatively analyze the spatial structure and non-spatial structure of the ancient tea tree (*Camellia taliensis*) community in Qianjiazhai, Ailao Mountain. The main conclusions are summarized below.

BLS is a high-efficiency and non-destructive tool for obtaining structural information on both common and rare species. Compared with traditional methods, BLS not only requires considerably less work and time but also helps to minimize interference with the environment. The ancient tea tree community exhibited a clustered spatial pattern, uniform tree size, and strong interspecific isolation. Competitive pressure was most intense among ancient tea trees in diameter class I (5–15 cm), and it gradually decreased with increasing diameter class. The community demonstrated rich species composition, high species diversity, and a relatively balanced distribution of individuals among species, with the ancient tea tree as the dominant species. The diameter distribution followed an inverse J-shaped curve, indicating strong regeneration ability and significant potential for sustained natural succession. These characteristics indicate that the function and structure of the ancient tea tree community were relatively stable.

## Figures and Tables

**Figure 1 plants-14-02485-f001:**
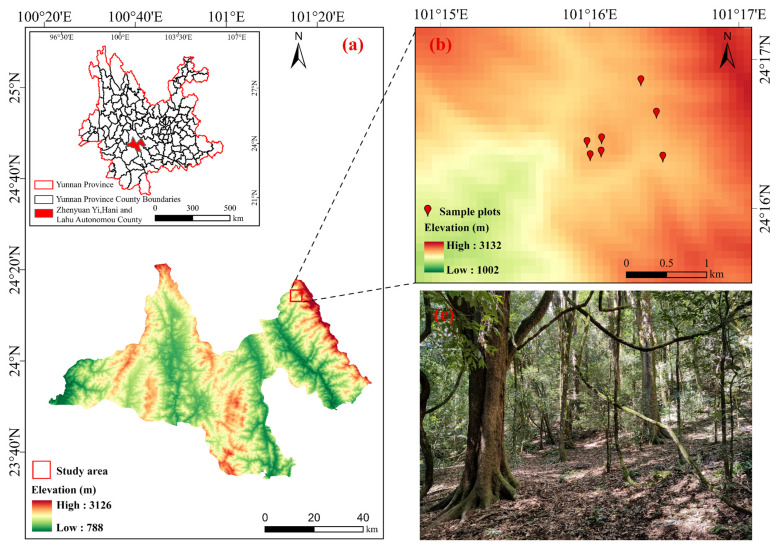
Location of the study area and location of sample plots. (**a**) Location of Zhenyuan Yi, Hani, and Lahu Autonomous County in the Yunnan province; (**b**) distribution of 7 sample plots; (**c**) real-life image of one ancient tea tree sample plot.

**Figure 2 plants-14-02485-f002:**
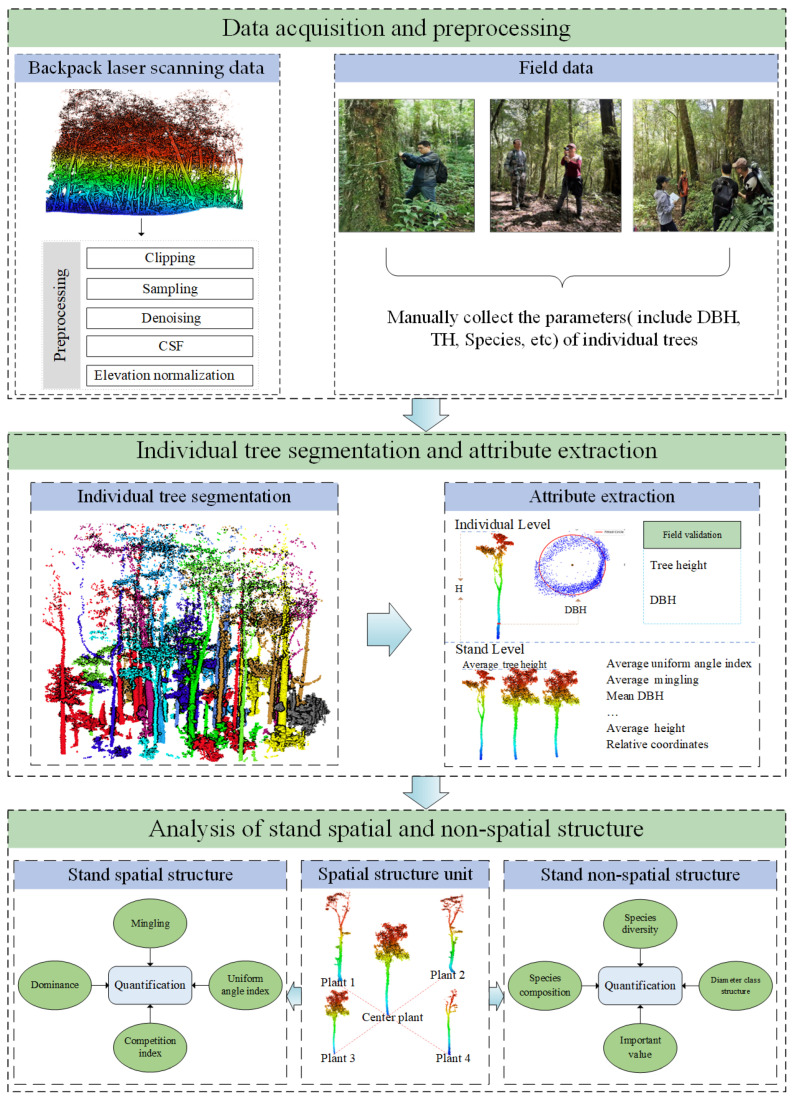
Workflow of research.

**Figure 3 plants-14-02485-f003:**
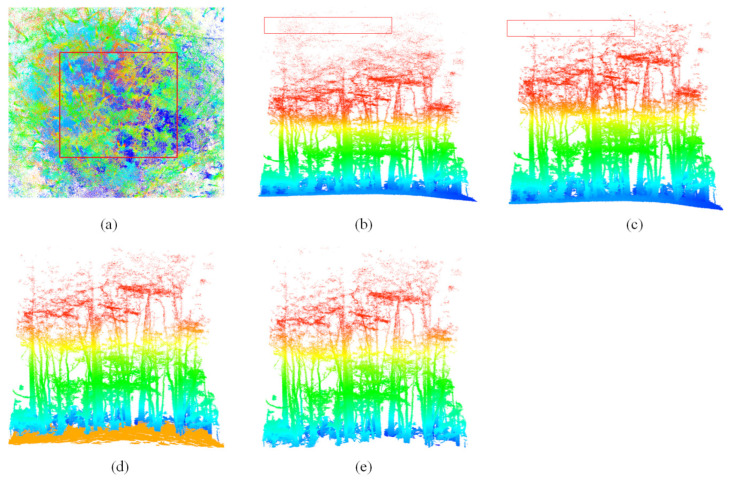
Pre-processing: (**a**) raw point cloud image before clipping, with the red rectangle indicating the actual sample plot; (**b**) point cloud image after clipping; (**c**) point cloud image after denoising and sampling; (**d**) ground and non-ground point cloud after CSF; (**e**) normalized results. Notes: The red rectangle in (**b**) represents the noise. The red rectangle in (**c**) represents the noise removed. The image in (**a**) shows a vertical view, and the other images show a side view.

**Figure 4 plants-14-02485-f004:**
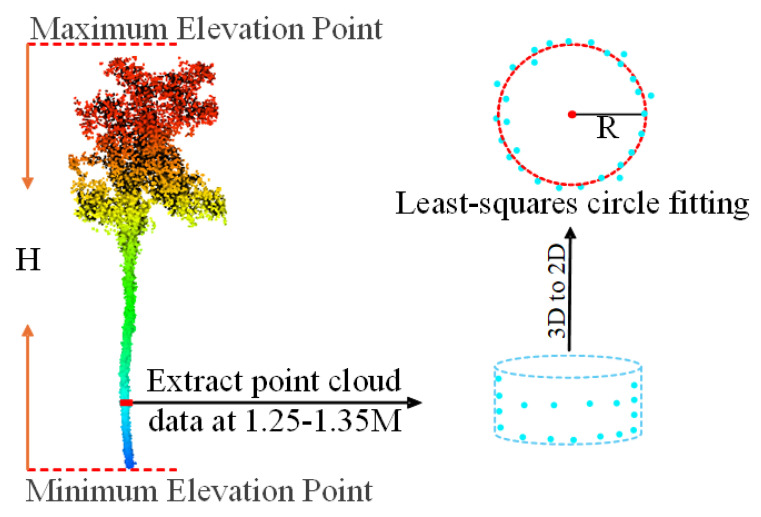
Extraction of DBH and TH processing.

**Figure 5 plants-14-02485-f005:**
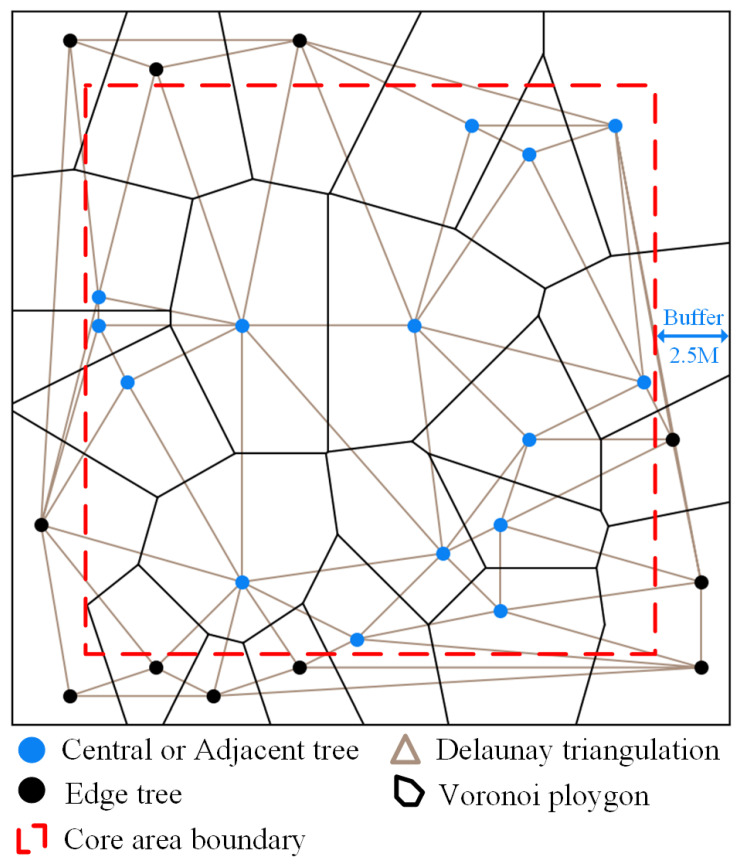
Illustration of delineating spatial structure units using Voronoi diagram.

**Figure 6 plants-14-02485-f006:**
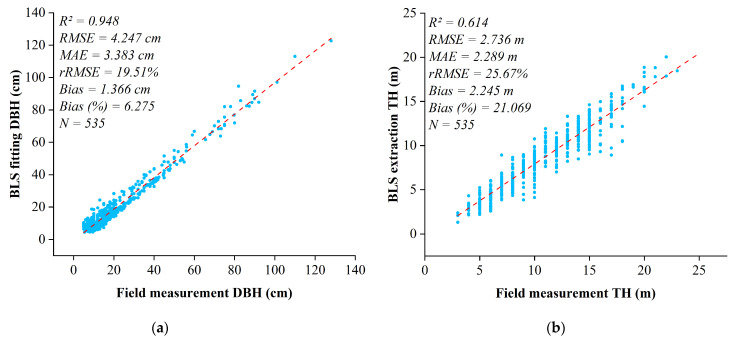
Comparisons of field-measured DBH and TH with DBH and TH extracted from BLS. (**a**) DBH extraction comparison; (**b**) TH extraction comparison.

**Figure 7 plants-14-02485-f007:**
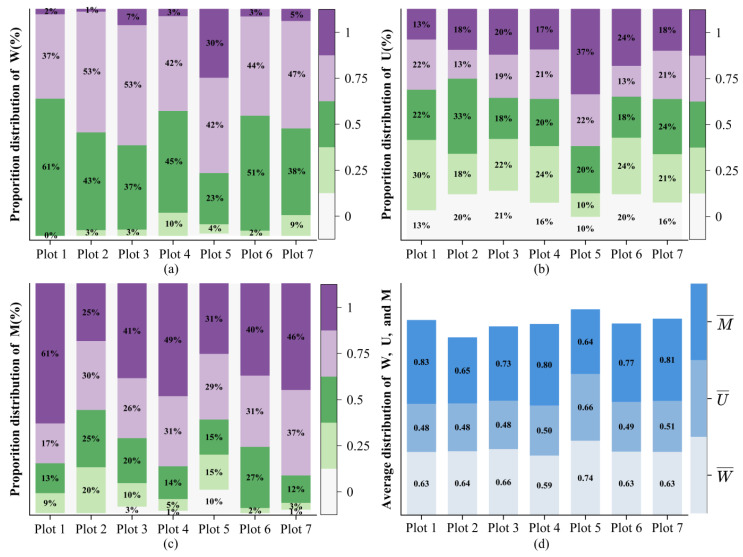
The distribution of uniform angle (W), dominance (U), and mingling (M) at different grades. (**a**) presents the distribution uniform angle at different grades. (**b**) presents the distribution of dominance at different grades. (**c**) presents the distribution of mingling at different grades. (**d**) presents the distribution of average uniform angle, average dominance, and average mingling at different sample plots.

**Figure 8 plants-14-02485-f008:**
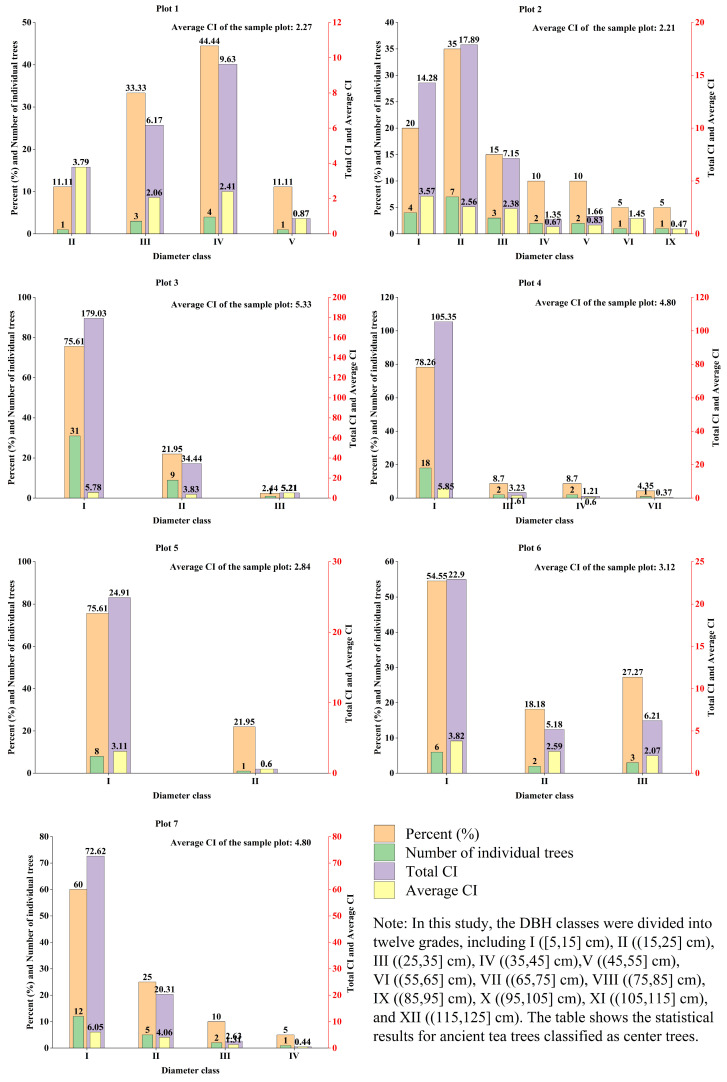
The distribution of ancient tea diameters and the competitive pressure in each individual sample plot.

**Figure 9 plants-14-02485-f009:**
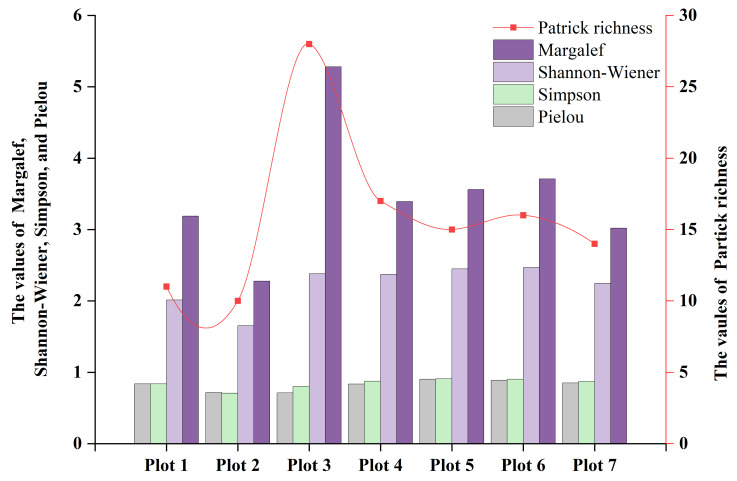
The distribution of species diversity indices for each individual sample plot.

**Figure 10 plants-14-02485-f010:**
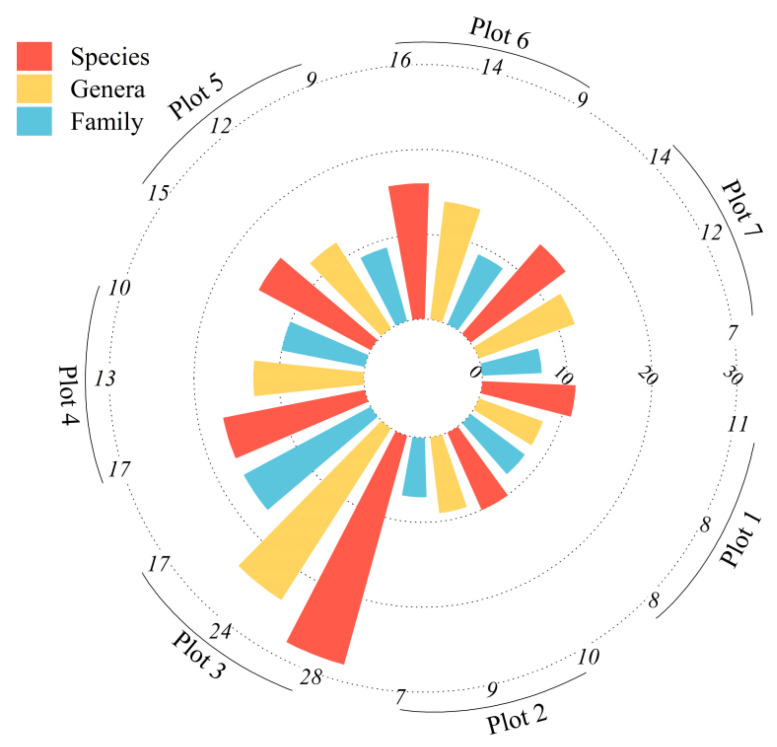
The distribution of families, genera, and species in each individual sample plot.

**Figure 11 plants-14-02485-f011:**
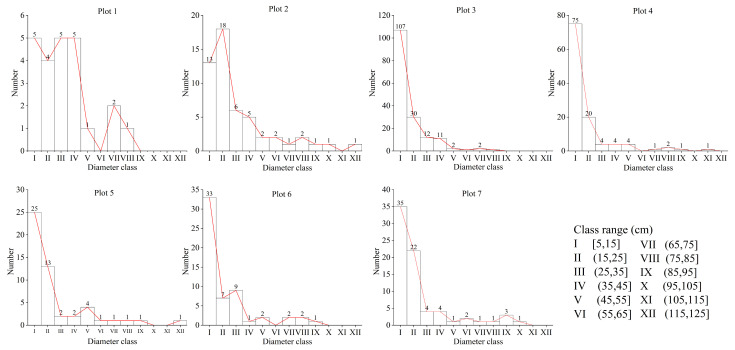
The distribution of DBH for each individual sample plot.

**Table 1 plants-14-02485-t001:** Basic information on sample plots.

Sample Plots	Number of Individual Trees	Average DBH (cm)	Average TH (m)	Max DBH (cm)	Min DBH (cm)	Max TH (m)	Min TH (m)	Slope (°)	Aspect	DEM (m)
Plot 1	23	33.41	12.88	82.34	8.97	20.04	3.86	15	NW	(2390–2399)
Plot 2	52	31.52	11.14	118.20	8.00	16.89	3.87	20	NW	(2298–2308)
Plot 3	166	16.08	7.59	82.48	5.00	14.37	1.31	15	SW	(2258–2262)
Plot 4	112	18.37	7.40	113.06	6.24	21.89	2.20	15	NW	(2217–2241)
Plot 5	51	24.57	9.20	122.62	6.00	16.61	3.21	18	NW	(2217–2241)
Plot 6	57	23.25	8.14	85.76	5.70	20.80	2.93	20	NW	(2217–2241)
Plot 7	74	23.27	8.02	95.40	5.92	18.70	2.15	21	NW	(2217–2241)

**Table 2 plants-14-02485-t002:** The distribution of the top 3 species based on importance values (IVs) in each individual sample plot.

Sample Plots	Species	Number of Individuals	Relative Abundance (%)	Relative Frequency (%)	Relative Dominance (%)	Important Value (%)
Plot 1	*Camellia taliensis*	9	39.13	31.58	40.30	37.00
	*Acer flabellatum*	2	8.70	10.53	20.20	13.14
	*Prunus undulata*	3	13.04	10.53	10.31	11.29
Plot 2	*Camellia taliensis*	27	51.92	41.67	53.42	49.00
	*Symplocos ramosissima*	7	13.46	11.11	10.44	11.67
	*Lithocarpus xylocarpus*	3	5.77	8.33	12.10	8.73
Plot 3	*Camellia taliensis*	70	42.17	25.58	33.93	33.89
	*Manglietia insignis*	14	8.43	8.14	8.20	8.26
	*Michelia floribunda*	11	6.63	6.98	6.89	6.83
Plot 4	*Camellia taliensis*	29	25.89	25.89	24.02	23.66
	*Prunus undulata*	19	16.96	16.96	18.08	16.94
	*Actinodaphne forrestii*	11	9.82	9.82	10.61	9.88
Plot 5	*Camellia taliensis*	11	21.57	18.60	17.90	19.36
	*Machilus yunnanensis*	7	13.73	11.63	5.87	10.41
	*Lithocarpus xylocarpus*	3	5.88	6.98	17.79	10.21
Plot 6	*Camellia taliensis*	14	24.56	17.07	22.61	21.41
	*Lithocarpus xylocarpus*	5	8.77	9.76	19.65	12.73
	*Prunus undulata*	7	12.28	12.20	7.37	10.61
Plot 7	*Camellia taliensis*	20	27.03	20.00	17.46	21.50
	*Manglietia insignis*	8	10.81	10.91	25.35	15.69
	*Machilus yunnanensis*	11	14.86	12.73	9.57	12.39

## Data Availability

The authors do not have permission to share data.

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
