# Peer review of "Stand Structure Extraction and Analysis of Camellia taliensis Communities in Qianjiazhai, Ailao Mountain, China, Based on Backpack Laser Scanning"

_plants, 2025, doi:10.3390/plants14162485_

Round 1
Reviewer 1 Report
Comments and Suggestions for Authors
The manuscript dealing with the analysis of stand structure of Camellia talinensis with remote sensing active sensors is well structured. However, materials and methods, results and discussion need further details and clarifications (see comments). Thus, major changes are recommended.
Comments
1) Figures quality need to be improved.
2) Line 26 – please consider replacing “diameter class I” by 5-15 cm diameter class.
3) Line 80 – please use the full scientific name of the species.
4) Line 99 – please consider using the scientific name of the species.
5) Table 1 – please consider including the diversity and competition indices (and units). Also, consider including Table 1 after the keywords and before the introduction.
6) Lines 221-228, 233-239, 294-303, 312-314 – please include the software used
7) Figure 5 - Voronoi polygons are difficult to see.
8) Lines 255-259 – please consider including references.
9) Line 278 – “diameter class structure” or diameter distribution.
10) Line 371-372 – please use always the same terminology. “Species richness” or Patrick richness.
11) Lines 385,395 – “top 3 species” or the three most frequent species?
12) Lines 385-394 – please check English.
13) Table 4 – “Individual number” or Number of individuals? Also, check the scientific names of the species, some are not according to the standard used.
14) Lines 408-410 – 9, 137 and 12 are number of individuals? Is so included it in the text.
15) Lines 410-412, Figure 10 – the J-shape diameter distribution is not clear in all plots. Also, the scale of the x axis can be misleading. Please provide further details in the text to justify the J-shape diameter distribution.
16) Lines 438-451 – please check English.
17) Lines45-460, 463-464 – please consider including references.
18) Lines 481-487, 517 - the J-shape diameter distribution is not clear in all plots. Moreover, the individuals of small diameter can be regeneration of mature dominated individuals. Please clarify.
Reviewer 2 Report
Comments and Suggestions for Authors
General comments:
Dear Authors,
I read with interest your article on the use of terrestrial remote sensing techniques to measure parameters used in broadly defined forestry, forest management, and environmental protection.
Your text clearly indicates that the method used has great potential in terms of accuracy and precision of measurement combined with efficient use of labor. I appreciate the use and adaptation of appropriate indicators, especially those related to biodiversity, in the development and analysis of the results.
The descriptions of methods, results with their illustrations, and discussion also deserve great appreciation.
After reviewing and addressing the comments I included in my review, I would like to recommend your article for publication.
- Backpack Laser Scanning vs. Mobile Laser Scanning.
Backpack Laser Scanning (BLS?) is a type of mobile laser scanning (MLS) – perhaps it's more appropriate to use the more general term "mobile laser scanning"? Especially since, for example, on page 2 (lines 63-67) you list three basic types of laser scanning, and Backpack Laser Scanning isn't among them.
- Advantages and disadvantages of MLS / BLS / TLS.
Page 2, line 67-71: It is worth noting, in addition to the above-mentioned advantages of TLS and MLS, that the higher data density (measurement points per unit area) results from the slower movement of the scanner (compared to aerial techniques and a smaller distance between the device and the tested surface. ), and has higher positioning precision.
Page 2, line 73-76: Primarily, due to the greater workload, TLS is used for single objects or relatively small areas, such as natural monuments (single trees) or landslides (in geomorphological studies).
- Ancient tea tree community.
You frequently use the term "ancient tea tree community" in your article (in the title, abstract, keywords, etc.). I'm not sure if the word "community" is appropriate here—perhaps a term like "grove", "wood" or "plantation (if it's a planting)" would be more appropriate.
- Study area.
The description of the research area should be placed first in the Materials and methods section (2.1.).
- Text formatting.
The last word of the sentence and the citation (number in square brackets) should be separated by a space each time.
A summary explanation of abbreviations used in an article is usually placed at the end of the text as an appendix, rather than as a table (page 3, table 1) in the introduction. A lot of shortcuts used in article is not included in this table (VI, indices’ abbreviations, RMSE, MAE, …).
Comments to the text:
- Page 1, line 41: Ancient tea tree community affects the stability and ecosystem function of the community – apart from my doubts about the word “community”, it is used twice in the sentence. Maybe “Ancient tea trees affect the stability and functions of its ecosystems”?
- Page 2, line 47: The richest ancient tea tree resources → the richest ancient tea tree habitats.
- Page 2, line 51: To minimize damage to the community → to minimize interference with the environment.
- Page 3, table 1: Some of the abbreviations used in the text (for example diversity index Importance Value / IV or Airborne Laser Scanning / ALS) are not listed in the table. Why you don’t use alphabetical order in the shortcuts list?
- Page 4, figure 1: Correct the description in the “Field data” box – there is no subject, the opening bracket is incorrectly separated by a space, there is no period next to “etc.”. I suggest the following description: field measurement DBH, TH, species, etc. for individual trees. Also how was measured location of individual trees for comparison with laser scanning? GPS mapping? Descirbe it in the text or in the figure.
- Page 5, line 154: Dominant plant population in this area → dominant species in ecosystem.
- Page 5, figure 2: Add contour map of China’s provinces with highlighted Yunnan Province. Shouldn’t “Zhenyuan” start with capital letter? Elevation color scale (hypsometric scale) on both maps (a) and (b) should be inverted: low elevation values in greens, middle in yellow, high in reds. On linear scale only two segments, not three in left side, and change the length of the segment in the contour map in part (a) from 580 km to 500 km, in map (b) from 1.28 km to 1.00 km.
- Page 5-6, line 163-173 and table 2: The number of individual trees measured in the field for each sample plot is also an important parameter.
- Page 7, figure 3: On the part (a) is a vertical view, and in other parts side view.
- Page 8, line 233-248: There is no information of the software and the tool you used to create Voronoi polygons and Delaunay triangular network. In previous steps this information was added (point cloud magic, LiDAR360).
- Page 9, figure 5: Typos in the legend: “Centeral” instead of “Central”; “Voronio polygon” instead of “Voronoi polygon”. The symbols of Voronoi polygon in legend and on the scheme are not the same.
- Page 9, line 251: You used more than one stand structure index (11!), so the title of this section should be “Stand structure indices”.
- Page 11, line 315: Result → Results.
- Page 11, line 317-318: Only here is there information on how many trees were examined, without providing information on the number of trees at individual sample points.
- Page 12, line 331-348: Again, there is no information about the sample size (number of trees) for individual sample plots.
- Page 13, figure 8: Typo in the diagram legend: “Patrick richess: instead of “Patrick richness”.
- Page 13, line 379: The title is wrong: It is not the distribution – simply “Diversity indices…”; it shows diversity indices for each individual sample plot, not for sample plots at all.
- Page 15, table 4: Why is “camellia talinensis” the only one listed in the table with a lowercase letter?
- Page 16, figure 10: Same as in figure 8 (page 13, line 379).
- Page 17, line 424-426: I ask again: is this the size of your sample plot? If so, please include it in the Materials and Methods section. If not, please include the size of the sample plots you used.
- Page 17, line 426-427: And how many time was consumed for field measurement of DBH, CT and so on?
- Page 17, line 430: No space between RANSAC and (Random Sample Consensus). In the table 1 the name is written in lowercase letters (Random sample consensus).
- Page 18, line 468: No citation in the brackets directly by names MacArthur and Elton, like e.g. in the line 475 (Tilman [71]).
- Page 18, line 499: Primarily on manual recognition → primarily on recognition during field works.
- Page 18, line 510-514: “The BLS is a high-efficiency and non-destructive tool for obtaining information on rare species” –this is a general statement, and why only obtain information on "rare" species? Is BLS useless for "common" species? In any case, this is not the conclusion from the article - write that, above all, you obtained comparable measurement results from BLS to field research, with significantly less work, time, and last but not least, less pressure on the studied plants.
- Page 19, table A 1 from Appendix A: Again, why is “camellia talinensis” the only one name listed in the table with a lowercase letter?
Reviewer 3 Report
Comments and Suggestions for Authors
General Comments
Dear authors, This study presents an analysis based on backpack laser scanning (BLS) to assess forest structure. The methodology is generally comprehensive and relies on established techniques for spatial structure analysis. The novelty of the study lies in the approach to field data acquisition using BLS and the application to a specific case study to derive spatial structure characteristics. The following comments aim to clarify and improve the manuscript to make it more precise, informative, and accessible to readers.Abstract
-
Lines 23–25: Please clarify the sentence:
"Secondly, this community exhibits a clustered spatial distribution (average uniform angle > 0.59), uniform tree sizes (average dominance > 0.48), high interspecific isolation (average mingling > 0.64)."
This sentence could be made clearer for readers who are not specialists in spatial structure metrics. -
Consider whether the following statement is overly optimistic:
"The results indicate that the structure and function of this community collectively exhibit excellent stability."
Long-term stability involves dynamic factors such as climate change and disease, which are not discussed in the manuscript. Please soften this conclusion or provide more evidence to support it. This was not included also in your Conclusion. - Indicate how your findings compare with similar ecosystems or previous research.
1. Introduction
- Line 40: Add spacing around citations, e.g., "value[1]" → "value [1]".
- Line 65: Consider including spaceborne laser scanning as a fourth category, following airborne LiDAR.
- The necessity of Table 1 listing abbreviations is unclear, especially if these abbreviations are not used extensively throughout the manuscript.
- In the final paragraph of the introduction, clearly state the aim and objectives of the study. Explicitly mention whether the work focuses on using BLS-derived data to analyze the stand structure of the Camellia talinensis community for ecological understanding.
2. Materials and Methods
- Please clarify the rationale behind selecting 7 sample plots. Would additional plots have changed the results significantly? Are you combining data from these plots or treating them separately?
- There is insufficient detail on forest management practices in the ancient tea tree community. Are there silvicultural interventions (e.g., pruning, thinning)? Is harvesting practiced?
- Line 158: In Fig. 2, the upper-right DEM — is the range in elevation across sample plots substantial?
- Line 174: Consider including the number of trees per plot in Table 2 to indicate stand density.
- Additionally, explain how you validated tree positions Was a ground-truthing dataset used?
- Lines 219–220: How were misclassified trees identified and manually corrected?
3. Results
- Line 317: Does the evaluation of BLS accuracy include all field data across plots?
- Line 328: In Fig. 6, did you analyze correlations for all trees from all the plots combined? If there are differences across plots, please describe them. Note that height measurements may be prone to error.
- Line 350: Instead of only abbreviations, please describe the contents of Fig. 7 more clearly, e.g.: The distribution of uniform angle index (W), U, M.
- Line 381: Add a more detailed description of Table 3 and explain its significance.
-
Line 354: Rephrase this unclear sentence: "Table 3 statistics the distribution of ancient tea DBH..."
Suggestion: "Table 3 presents the distribution of DBH ..."
Also, the table is quite difficult to interpret. Consider replacing it with a graphical visualization for better clarity. - Table 4: Capitalize the first letter of "Camellia".
- Line 440: You mention DBH values being overestimated or underestimated. Please consider adding relative bias metrics to quantify the accuracy of BLS-derived measurements.
- Line 414: Regarding Fig. 10, is the DBH distribution observed in your sample plots representative of the larger ecosystem or only representative to your case study area?
4. Discussion
- Line 427: You state that scanning one plot takes approximately 15 minutes per person. However, how long does data processing take? Please include also the processing time.
- Line 468: Provide reference number for MacArthur and Elton.
- Lines 492–493: Rephrase and clarify: "This method not only minimizes human disturbance during field investigations..." What kinds of disturbances are typically associated with classical fieldwork? Are these minimized with BLS? Please be specific.
5. Conclusions
- Line 510–511: "BLS is a high-efficiency and non-destructive tool for obtaining structural information on rare species." Consider rewording: BLS can be applied broadly and is not limited to rare species. BLS is suitable tool of both common and rare species.
- Line 519: Why do you state that the ancient tea tree community is “relatively stable”? Based on what metrics or th resholds? Clarify your reasoning.
- Line 508. Camellia taliensis or Camellia talinensis?
6. Patents
- Line 521: The section title "6. Patents" is included, but no content follows. Did you intend to report a patent? If not, remove the heading.
Round 2
Reviewer 1 Report
Comments and Suggestions for Authors
The authors answer to most of the reviewer comments, yet there are still two issues that should be addressed (see comments). Minor changes are recommended.
Comments
1) Line 86 – Please replace “Metasequoia plantation forest” by Metasequoia glyptostroboides plantation forest.
2) Table A 1 - please consider including the units of the diversity and competition indices
Reviewer 2 Report
Comments and Suggestions for Authors
Dear Authors,
I see that you have significantly improved your text, which has increased its scientific value. I recommend it for publication in the Plants journal.
Editing note: one paragraph (page 12, lines 343-354) is not justified.
Reviewer 3 Report
Comments and Suggestions for Authors
Dear Authors,
Thank you for your careful consideration of my initial comments and your thorough point-by-point responses. Methodologically, this paper is scientifically sound and presents valuable insights into the structure of ancient tea tree communities.
However, I would like to raise a few remaining concerns based on Comment/Response 1 and 8 (Materials and Methods) and Comment/Response 1 and 3 (Discussion) that I believe would further strengthen the manuscript:
1. Sampling Fraction and Inference Scale:
Please consider including the total area of the case study site to allow readers to evaluate the sampling fraction. This addition would help clarify the reliability and limits of inference, particularly when drawing broader conclusions from seven sample plots. A clear connection between sampling intensity and inference scale is essential for contextualizing your results.
2. Processing Efficiency:
The current data processing time—especially for pre-processing and manual corrections—appears too time-intensive for practical application in larger-scale studies. Please estimate the total processing time of all plots, and briefly discuss the implications for operational scalability. Suggestions for how to improve efficiency would be welcome.
3. Team Size and Field Disturbance:
In your description of BLS (Backpack Laser Scanning), please clarify the actual field team size required for efficient data collection under typical field conditions. While BLS reduces certain types of ecological disturbance, traditional field methods often require 2–3 people but may not always result in significant ecological impacts, depending on site sensitivity. Avoid overemphasizing the drawbacks of conventional methods unless they are well-documented in your study context. A balanced discussion of disturbance trade-offs between traditional and BLS methods would improve objectivity.
The previous responses:
Comment 1: Please clarify the rationale behind selecting 7 sample plots. Would additional plots have changed the results significantly? Are you combining data from these plots or treating them separately?
Response 1: Many thanks to you for your comments. According to the distribution of ancient tea trees of the study area, 7 representative sample plots of ancient tea tree communities were selected. Given that these 7 sample plots are sufficiently representative, adding additional plots would not produce statistically significant differences in the results. The the data of 7 sample plots were treated separately in data processing, but we combined data of these plots for results analysis.
Comment 8: Line 414: Regarding Fig. 10, is the DBH distribution observed in your sample plots representative of the larger ecosystem or only representative to your case study area?
Response 8: Many thanks to you for your comments. The sample plots used in this study were selected from the most representative areas of the ancient tea tree community in Qianjiazhai. Therefore, the DBH distribution shown in Fig. 11 reflects only the structural characteristics of ancient tea trees in Qianjiazhai and may not be representative of other communities beyond the study area.
Discussion
Comment 1: Line 427: You state that scanning one plot takes approximately 15 minutes per person. However, how long does data processing take? Please include also the processing time.
Response 1: Many thanks to you for your comments. We have added the data processing time in the revised manuscript (see Line 463-468).
(“In this study, the average time for BLS to collect data from each plot (25 m x 25 m) was approximately 15 minutes. Due to the rugged terrain of the study area, field data collection was limited to an average of 1-2 sample plots per day. For each plot, BLS data pre-processing and individual tree segmentation took approximately 15 minutes, whereas correcting segmentation errors (such as misdivided trees) required an additional average of 1-2 hours.”)
Comment 3: Lines 492–493: Rephrase and clarify: "This method not only minimizes human disturbance during field investigations..." What kinds of disturbances are typically associated with classical fieldwork? Are these minimized with BLS? Please be specific.
Response 3: Many thanks to you for your comments. We have revised the relevant sentence to clarify the kinds of disturbances typically associated with traditional forestry surveys, such as vegetation trampling, soil disturbance, and ecological degradation—especially in fragile ecosystems. We also explained that as a portable, non-contact 3D scanning technology, BLS allows efficient data acquisition with minimal field intervention, thus substantially reducing ecological disturbance (see Line 544-549). The revised text now reads:
(Traditional forestry surveys often require numerous personnel to work within field plots, which can cause vegetation trampling, soil disturbance, and ecological degradation, with particularly severe impacts in fragile ecosystems [86]. As a portable, non-contact three-dimensional scanning technology, BLS enables efficient data acquisition with minimal field intervention, thereby substantially reducing ecological disturbance [15].)
